# The Contagion of Psychopathology across Different Psychiatric Disorders: A Comparative Theoretical Analysis

**DOI:** 10.3390/brainsci12010067

**Published:** 2021-12-31

**Authors:** Danny Horesh, Ilanit Hasson-Ohayon, Anna Harwood-Gross

**Affiliations:** 1Department of Psychology, Bar-Ilan University, Ramat Gan 5290002, Israel; ilanit.hasson-ohayon@biu.ac.il (I.H.-O.); ANNAjudithharwood@gmail.com (A.H.-G.); 2Department of Psychiatry, NYU Grossman School of Medicine, New York, NY 10016, USA

**Keywords:** emotional contagion, psychopathology, transdiagnostic, PTSD, depression, OCD, psychosis

## Abstract

Psychopathology is often studied and treated from an individual-centered approach. However, studies have shown that psychological distress is often best understood from a contextual, environmental perspective. This paper explores the literature on emotional contagion and symptom transmission in psychopathology, i.e., the complex ways in which one person’s psychological distress may yield symptoms among others in his/her close environment. We argue that emotions, cognitions, and behaviors often do not stay within the borders of the individual, but rather represent intricate dynamic experiences that are shared by individuals, as well as transmitted between them. While this claim was comprehensively studied in the context of some disorders (e.g., secondary traumatization and the “mimicking” of symptoms among those close to a trauma survivor), it was very scarcely examined in the context of others. We aim to bridge this gap in knowledge by examining the literature on symptom transmission across four distinct psychiatric disorders: PTSD, major depression, OCD, and psychosis. We first review the literature on emotional contagion in each disorder separately, and then we subsequently conduct a comparative analysis highlighting the shared and differential mechanisms underlying these processes in all four disorders. In this era of transdiagnostic conceptualizations of psychopathology, such an examination is timely, and it may carry important clinical implications.

## 1. Introduction

Psychopathology is most often studied, and treated, from an individual-centered approach. Ever since Freud conceptualized neurosis and its complex psychodynamics [1], psychology has become a largely phenomenological discipline, with a unique emphasis on one’s subjective and deeply personal experience of psychic pain. Nonetheless, it has quickly become apparent that no individual lives outside of a broader human context, inhabited by parents, children, spouses and one’s community as a whole. This realization, which became present in both the psychoanalytic [2] and empirical [3] literature, was subsequently the advent for the arrival of systemic approaches, family psychology theories, and contextual models of psychology, all of which have acknowledged psychological processes shared by more than one individual.

This paper is an attempt to explore what may be termed “shared experiences” in psychopathology. We refer to “shared experiences” as a broad umbrella for a variety of similar constructs, such as “transmission of psychopathology” and “emotional contagion”, that are often used and studied in different contexts. We argue that emotions, cognitions and behaviors are not strictly intrinsic entities which stay within the borders of the individual, but rather represent complex dynamic experiences that may be shared by individuals, as well as transmitted between them. While this claim was comprehensively studied in the context of some psychiatric disorders (e.g., PTSD), it was very scarcely examined in the context of others. We aim to bridge this gap in knowledge, by attempting to identify the existence of symptom transmission and its unique and differential mechanisms, across four distinct psychiatric disorders: post-traumatic stress disorder (PTSD), major depressive disorder (MDD), obsessive-compulsive disorder (OCD), and psychosis. 

The rationale for selecting these specific four disorders for comparison was two-fold. First, they belong to presumably four distinct classes of psychopathology according to the DSM-5 [4]: stress-related (PTSD), mood disorders (MDD), obsessive compulsive, and psychosis. Second, we identified a sufficient amount of literature on symptom transmission in each disorder, this enabling a meaningful theoretical analysis. By reviewing the literature on symptom transmission on each of the four disorders, we hope to achieve a transdiagnostic understanding of this phenomenon. This comparative analysis is guided by two leading questions: (1). Which aspects of the psychopathological experience are transmitted (emotions, cognitions, and behaviors)? (2). Which factors facilitate or buffer transmission, that is, what are the underlying mechanisms of symptom contagion?

Notably, several important considerations directed the literature search. First, we focus mainly on the psychological aspects of transmission, with only limited discussion of genetics and epigenetics, which are beyond the scope of this paper. Accordingly, we focus on the broad concept of emotional contagion, as well as on more specific concepts, such as secondary trauma in the case of PTSD. Second, throughout this paper, we wish to distinguish between what may be referred to as general caregiver burden and more explicit processes of symptom transmission. Notably, sharing the experience of psychopathology and family/caregiver burden were not always easy to differentiate as the two may interact. Finally, in our analysis we rely on various sources of knowledge, including empirical studies, theoretical discussions and case reports. We begin our review with one of the most well-known conceptualizations of psychopathological transmission—emotional contagion.

## 2. Emotional Contagion and Related Constructs

Emotion, most simply defined as “a feeling that is often short-lived, intense, and specific” [5], is a phenomenological experience. Although it has been traditionally contrasted with reason, today it is often understood as comprising a complex interaction of cognitions, behaviors, and feelings, as well as neuropsychological changes [6]. Emotions are tightly woven into every aspect of human experience [6], having both universal and culturally-specific elements [7]. While emotions are most often studied at an individual level, scholars have identified group-shared and group-based emotions, which involve the convergence of emotions across members of a certain social group [8]. The term “emotional contagion” metaphorically refers to “catching” the emotions of others, as one would with a physical disease. It is a multilevel phenomenon, in which emotions generated by one individual are met with emotional, attentional, or behavioral synchrony in another individual, presumably in a largely unconscious manner [9,10].

The contagious nature of emotions can be traced back to the origins of crowd psychology and the likening of the spread of emotions, such as fear and anger, to the spread of germs or diseases [11]. This notion was further elucidated in the early studies of psychoanalysis [12] and social psychology, with Becker [13] referring to the phenomenon of “mimpathy”, or the unconscious imitation of another’s emotional state. For this theoretical analysis, it is important to provide an accurate definition of contagion, as the specific boundaries of this concept are often blurred. The contagion model attempts to explain the transmission of emotions from one person, who is the primary emotional “carrier”, to another person, who subsequently becomes a “secondary” carrier [14]. This description draws much from the physical-medical realm. In addition, feeling emotional pain as a result of being in the presence of another’s emotional pain would not be strictly considered contagion although some scholars do tend to include such processers under its very broad umbrella. Finally, it should be noted that emotional contagion has been demonstrated in the context of both negative [15] and positive [16] emotions. Susceptibility to such contagion may be influenced by the individual’s psychological makeup. Numerous factors were identified to play a role in the transmission of psychopathology, many of which are addressed in the following sections. 

Emotional contagion appears to have evolutionary origins. This notion is supported by research into neural interactions between two “brains” [17] and the neural consequences of viewing behaviors or states in another. The discovery of mirror neurons [18] further contributed to the understanding of how the perception of a behavior in another activates the viewer’s own corresponding neural representations as if they were performing the action themselves. Furthermore, emotional contagion seems to appear in non-humans as well [19], further indicating that it is a broad, highly common process. Interestingly, emotional contagion may occur through social appraisal [20], a process which is built on the theory that people take information from the behaviors of those around them when determining their own emotional state [21]. This has been evidenced with anxiety and fear being mediated by the reactions of others present in the situation [22]. While social appraisal implies a cognitive, conscious analysis of the situation, it could interact with more automatic mimicking responses related to the “caught” emotion [20]. Finally, while this paper focuses on contagion of symptoms in psychiatric disorders, it should be noted that contagion has been relatively broadly studied among both non-clinical [23] and subsyndromal or preclinical samples [24]. Thus, it seems to be a broad phenomenon, appearing among a wide variety of populations.

## 3. Symptom Contagion in Four Psychiatric Disorders

While emotional contagion thus far has been mostly described in reference to everyday emotions (e.g., loneliness, and happiness) likely to be experienced by both the “infecting” and “infected” parties, its study has been extended to various psychopathologies as well. Due to the powerful, often debilitating nature of psychiatric disorders, it may be expected that the emotions, cognitions, and behaviors associated with them will have a significant effect on one’s environment. 

The terminology referring to the transmission of psychopathology has varied widely. In addition to the descriptions of contagion of emotional experiences, other terms describing similar phenomena have been used in the psychological literature. For example, the term “Folie a Deux”, as coined by Lasegue and Falret in 1877 [25], has been commonly used to delineate the transmission or contagion of psychotic symptoms, with over a hundred case studies published in the scientific literature using this terminology. As will also be discussed, in the field of post-traumatic stress, several terms have been interchangeably used to describe the infectious nature of psychopathology. Conducting a theoretical analysis on this topic therefore demands a broad usage of synonyms including “transmission”, “spread”, “mimicking of symptoms” or “psychopathology”, as well as the study of secondary symptoms in partners and family members. In this paper, we review the literature on these and other concepts, in order to expand our understanding of these processes in different psychopathologies. In order to achieve a transdiagnostic understanding of symptom contagion, one first needs to understand how this type of contagion is manifested in different disorders. Then, it would be possible to identify shared and differential processes between disorders, leading to a more integrative psychopathological perspective. We begin our review in the field of trauma, where the broadest body of knowledge on symptom transmission currently exists.

### Post-Traumatic Stress Disorder (PTSD) and the Transmission of Traumatic Stress 

Post-Traumatic Stress Disorder (PTSD) is the most common chronic stress disorder, resulting from exposure to traumatic events. In the DSM-5 [4], diagnostic criteria for PTSD are as follows: First, an exposure (direct or indirect) to actual or threatened death, serious injury, or sexual violence must have taken place. In addition, there are four main symptom clusters: intrusive thoughts and re-experiencing of the trauma; cognitive and behavioral avoidance of reminders associated with the traumatic event; negative alterations in cognition and mood, and hyper-arousal. It has by now been well-established that trauma also exerts its influence beyond PTSD. Individuals exposed to traumatic events have been continuously found to suffer from a wide variety of symptoms and difficulties, including major depression; substance abuse; anxiety disorders; somatization [26]; and a myriad of marital, familial and parental difficulties [27].

Nowhere has the transmission of symptoms been more widely studied than in the context of traumatic stress. Looking at the historical evolution of trauma studies, one may notice a gradual shift from research concentrating on the so-called “primary” trauma victim, to more systemic examinations of post-traumatic couples, families, communities, and even nations [28]. Over the last three decades, studies have shown that individuals in the trauma survivor’s close proximity may also suffer from psychological symptoms associated with his/her experience. Thus, those surrounding the survivor may develop psychological and emotional difficulties, even if they were not directly exposed to the trauma [29]. In fact, this notion has recently received formal recognition in the DSM-5, which, for the first time in its history, included indirect exposure as a potential source of post-traumatic symptoms [30].

Different terms have been used to describe post-traumatic symptom transmission, with different emphases. While they are often used interchangeably, each term does carry its own specific connotation and context. “Secondary traumatization” [31] usually represents a general term, referring to the effects of trauma on one’s close environment. This term often serves as a broad conceptual umbrella, containing some of the other terms, described below. Most of these terms refer to what may be defined as “the cost of caring” [32]. They therefore mainly apply to therapists, aid workers and caregivers, all of whom are often deeply involved with the trauma victim in an attempt to assist and alleviate his/her distress. In the 1980s, Charles Figley presented what has since become the classic formative theory of secondary traumatization. In his original conceptualization, the symptoms of secondary traumatization are similar to those observed among the direct trauma victim, and may include nightmares, insomnia, loss of interest, irritability, chronic fatigue and changes in self-perception. This process is often thought to be associated with feelings of identification and empathy towards the traumatic experiences of a loved one. According to Figley [33] the process of secondary traumatization starts with one’s efforts to emotionally support one’s traumatized loved one, which, in turn, lead to ongoing attempts to empathize and understand the other person’s feelings and experiences. In the process of gathering information about their suffering, significant others might take on the traumatized person’s feelings, experiences, memories, and distress, as their own. Figley [34] later referred to the process of “compassion fatigue”, which mostly occurs among therapists and caregivers, and is manifested in feelings of faintness, confusion, isolation from friends and relatives, and a general sense of psychological depletion. As noted above, other related conceptualizations have also appeared over the years. McCann and Pearlman [35], who coined the term “vicarious traumatization”, advocate the “infection model”. The authors argue that the patient’s post-traumatic symptoms may infect the therapist. They particularly highlight symptoms of depression, cynicism, boredom and loss of empathy. Finally, the term “burnout” is also sometimes used to describe the psychological cost of caring, particularly among professionals such as nurses, police officers, firemen and correctional facility guards, all of whom may be vicariously exposed to traumatic circumstances [36,37].

Early studies in the area of secondary traumatization have focused on Holocaust survivors and their families, often showing a relatively heavy burden of secondary post-traumatic distress among individuals who were second generation to the Holocaust [38]. Over the past few decades, trauma transmission has been documented among a variety of other populations, including wives of combat veterans [39], mental-health professionals [40], and medical personnel [41]. Nonetheless, when examining the vast literature, one must differentiate between several perspectives about what constitutes “transmission”. The first definition has to do with instances in which individuals who were only indirectly exposed to the traumatic event developed PTSD symptoms. These are clear instances of secondary traumatization, where one’s symptoms are specifically connected to another person’s trauma, i.e., flashbacks related to the primary victim’s experience, avoidance of specific cues associated with the victim’s trauma, and vivid “memories” of events that occurred in the primary victim’s life [42]. A different perspective refers to more generalized distress in people who have a close relationship with the trauma survivor. This perspective moves beyond PTSD and comorbid psychiatric symptoms to include a wide range of other manifestations of distress and dysfunction, including marital, familial and inter-personal problems [43].

Most studies in the area of secondary traumatization have aimed to identify factors associated with risk and resilience to this phenomenon. These include higher levels of empathy, insecure attachment, a fusion style of self-differentiation, and female gender [44]. Other factors are related to the primary trauma victim, with most research indicating that higher levels of PTSD may entail more symptom transmission [42]. In line with Figley’s original conceptualization, some of these studies have focused on factors related to the regulation of interpersonal intimacy and distance, assuming that those who fail to protect themselves from over-involvement with the primary victim may find themselves carrying the wounds of trauma themselves. While some argue that a tendency for increased empathy among those in the trauma victim’s proximity may render them more vulnerable to post-traumatic distress [45], others have shown empathy’s protective role vis-à-vis secondary traumatization [46]. Recent studies have found empathy to be a more multidimensional construct. According to this view, there are several types of empathy, with each potentially leading to different psychological outcomes in the face of secondary exposure to trauma. For example, Dekel and colleagues [47] have differentiated between the cognitive, and possibly more positive aspects of empathy, and its more emotional aspects, which may be associated with secondary psychopathology. 

Several studies have focused on the role of another interpersonal factor in secondary traumatization: differentiation of self (i.e., one’s ability to remain close to another person, while still maintaining a sense of separation and self-identity). Perhaps not surprisingly, most studies [44,48] show that the extreme ends of differentiation, i.e., being enmeshed with the trauma victim, or cutoff from him/her, may serve as a risk factor for secondary traumatization, as both represent maladaptive interpersonal patterns of being either too close (fusion) to the distress of others or actively avoiding it (cutoff). Both poles eventually may lead to the same pathology, as they represent dysregulated interpersonal relationships. For example, when one attempts to distance oneself from the distressed individual, this may not serve to protect the former, but rather increase worry and guilt. This may also occur when the distressed individual is the one pushing close others away, or avoiding their presence, thus leaving them helpless and worried. Finally, attachment style was another factor studied vis-à-vis secondary traumatization. For example, studies have shown that individuals with an anxious attachment style may feel emotionally overwhelmed by their traumatized partner’s distress and over-identify with his/her experience [49].

Other factors associated with secondary traumatization have also been suggested. In the dyadic sphere, one of the most well-known hypotheses is that of “Assortative Mating”, according to which people who are in some fundamental way (e.g., genetics, temperament, and personality) similar to each other may be drawn together to form a couple. However, this explanation has been very scarcely studied, with the few trauma studies that were conducted failing to support it [50]. Female gender was also found to be a risk factor for secondary PTSD. According to a meta-analysis on gender differences in secondary traumatization [51], females showed higher susceptibility to this phenomenon across a variety of studies. This is in line with females’ more general well-established vulnerability to post-traumatic distress, which may be attributed to a variety of cognitive, emotional and biological factors [52]. However, women’s seemingly increased vulnerability may also be attributed to factors with a specific relevance to secondary traumatic stress, such as their basic socialization to be nurturing and emotionally available [10].

Other factors associated with risk and resilience to secondary traumatic stress have been studied specifically in relation to trauma therapists’ compassion fatigue, i.e., the distress professionals may experience when facing their clients’ pain and distress. For example, studies have shown younger trauma therapists to be more vulnerable to distress, while therapists who were more experienced, and/or applied more evidence-based practices, were more resilient to compassion fatigue [53]. However, evidence in this area is inconclusive. This can be seen, for example, when reviewing the mixed findings regarding the role of supervision quality and supervision satisfaction in therapists’ compassion fatigue [54].

Interestingly, the transmission of post-traumatic symptoms and distress has been studied not only on the interpersonal level, but also on a more macro-level. For example, studies assessing PTSD among residents of inner-city US neighborhoods have shown high rates of post-traumatic distress, which may be explained, in part, by an unusually high rate of vicarious trauma exposure. In areas where members of entire extended families often live only a few blocks away from each other, hearing about a loved one’s traumatic experience may entail feelings of fear and horror among those who were not directly exposed [55]. Furthermore, recent studies of PTSD following natural disasters have introduced novel methodologies to model the spread of post-traumatic symptoms across geographical areas. For example, a study conducted following the 2013 Hurricane Sandy in New York City [56] has shed important light on the ways in which post-traumatic symptoms may become concentrated, or “clustered”, in certain areas. Results indicated a spatial variation in risk of psychopathology within and across NYC boroughs. These findings are in line with more general models of emotional contagion, according to which symptoms tend to spread and cluster together in one’s environment [57]. 

## 4. Depression

Major depression is classified as a prolonged period of almost daily depressed mood; anhedonia; and other associated cognitive and behavioral symptoms, such as insomnia, lack of appetite, and inability to concentrate [4]. While genetic influences have been widely studied, the effects of environmental influences, childhood adversity, and a myriad of other factors highlight the complexity of the development of depressive symptomology [58,59].

Symptom contagion has been documented in major depression. Joiner and Katz’s [24] meta-analysis of depressive contagion included many laboratory studies of induced negative mood, which extended from physical interactions with depressed persons [60] to listening to tape recordings of depressed individuals [61]. Widely used negative-mood induction procedures include watching and listening to clips of sad people [62], with participants subsequently reporting feeling emotions such as “sad”, “downhearted” and “distressed” [63]. Likewise, even just moderating the amount of positive and negative posts a viewer sees on his/her Facebook news feed has demonstrated a small but significant effect on viewers’ own positive or negative posts [64].

Disentangling the depression contagion effect from genetic and shared-environmental effects may prove to be quite challenging, as it appears to be present at multiple levels of inter-personal relationships. In a groundbreaking study of over 12,000 friends, family members and neighbors, it was demonstrated that a person is 93% more likely to be depressed if a person he/she is related to (one degree of separation) is depressed, dropping to 43% and 37% for second and third degrees of separation, respectively, and no effect once there are four or more degrees of separation from a depressed person [65]. This spread of depressive symptomology reflects the relative ease in which negative mood can be transmitted from one to another. Depression has been found to run in families [66], with children of depressed parents demonstrating a higher risk of depression even 20 years following childhood assessment [67]. Studies have suggested several explanations for this familial effect. First and foremost, a large and impressive body of the literature indicates the genetic nature of major depression [68]. Second, depressive symptoms within families, particularly as studied among parents and children, may be associated with a negative childrearing environment. Depression in mothers is linked to unresponsive parenting, a more negative perception of a child’s behaviors, and more negative parent-child interactions [69]. This negative environment may have a detrimental effect on children’s subsequent mental health [70]. Third, methodologically, many measures of childhood depression are completed by parents rather than the children themselves, leading to a suggested parental bias and potential reflection of the reporting parent’s own internal mental state [71].

Interestingly, studies have shown that, even when there was no genetic similarity (e.g., related and unrelated IVF pregnancies), and when parental warmth and hostility were controlled for, maternal depression was still a predictor of child depression [72]. Given that it appears that neither genetics nor parenting style, at least not in regard to levels of warmth and hostility expressed, were solely responsible for depressive symptoms among children to depressed parents, the potential effect of emotional contagion from parent to child may be considered. What causes this contagion, or at least this association between depressive symptomology in parent-child dyads, appears to be related to relationship style [73]. While children who show low levels of negative attachment cognitions (such as “I like to get my mother’s point of view on things I’m concerned about”) may demonstrate no significant fluctuation in their depression scores in relation to maternal depression scores amongst children with a high level of negative attachment cognitions (such as “My parents don’t understand what I am going through these days”), the fluctuations in depression scores appear to be in concurrence with their parents’ depression symptoms, further indicating some level of contagion [73]. 

Contagion of depression in childhood may also occur within child peer groups. While self-selection of close friends appears to follow an Assortative Mating hypotheses [74], over a four-year period peer depression was demonstrated to be a predictor of children’s own depression levels, regardless of their own initial depression levels. Fear of failure played a role in this effect [74]. Thus, when children experience high levels of social anxiety they appear to be especially susceptible to the effects of depressive symptoms expressed by their self-reported best friend, with the friend’s depressive symptoms predicting the subject’s depressive symptoms at a later time point [75,76]. This role of social anxiety may be understood in relation to the phenomenon of “co-rumination”, i.e., defined as extensively discussing and revisiting problems, speculating about problems, and focusing on negative feelings [77]. Social anxiety may lead to extensive ruminative conversations [78], which, in turn, have been shown to be associated with the contagion of depression amongst adolescents [79,80].

Studying socially unrelated dyads affords the opportunity to more carefully rule out alternative explanations regarding the transmission of depression. Thus, when completely unrelated and randomly assigned college roommates were examined at introduction and then again at three and six months following their cohabitation, ruminative thinking and hopelessness among the individual and his/her roommate were positively associated [81]. Increases in ruminative thinking and hopelessness were, in turn, related to increases in depression. Interestingly, however, in this case, the roommate’s initial depression levels did not appear to be contagious; rather, their cognitive style (such as how much they ruminated on negative thoughts) was the contagious element [81].

A third affected population is that of cohabiting romantic partners. Associations in depression levels between male and female spouses have been demonstrated extensively [82,83]. Longitudinal studies of depression have enabled to establish causality, by demonstrating the predictive role of one spouse’s depression on the other’s over time [84,85]. While, in children, co-rumination is considered a key mediator of the contagion process, in romantic couples, increased sharing of confidences and greater reception of support from the other predicted greater contagion of depression symptomatology from one partner to another. In a study by Tower and Kasl [85], both partners’ symptoms were tested sporadically over a six-year period. For wives in 1985 and for husbands in 1988, a spouse’s baseline depressive symptoms contributed independently to an increase in the other partner’s depression score. Interestingly, these findings were stronger when a couple was close. In a more recent study from Norway [86], couples were longitudinally assessed over time, during the perinatal events of pregnancy, childbirth, and early parenthood. Mothers’ depressive symptoms late in pregnancy predicted elevated symptom levels in fathers 6 weeks after birth, with a small effect size. When attachment style was entered as a moderator, more effects were found. Among parents characterized by insecure partner-attachment styles, additional cross-lagged pathways of depression were evident during pregnancy and throughout the first year of parenthood. 

As in the aforementioned field of trauma, the caregiver burden has also been studied in relation to depression. For example, caring for a partner with physical health problems can place a high psychological burden on the unaffected partner, increase their own illness-related depression and, most important, demonstrate a correlation with the ill partner’s increasing depression [84]. However, this line of research poses a challenge to researchers, as it is often difficult to differentiate between purely contagious processes and more general processes related to feelings of burden, fatigue and burnout. As was noted above, being in a relationship with a spouse who copes with psychological difficulties often places an increased strain on the partner, so while an increase in depression symptoms among the partner over time may indicate the contagious nature of depression [87], so too may it indicate the increased burden presented by the sufferer [88]. A portion of this burden may be due to the more negative communication styles characterizing the depressed partner, leading to an overall more negative experience for both the depressed and non-depressed partner [89]. Finally, as was the case with PTSD, this relationship is most often seen in wives as opposed to husbands [83].

Understanding the mechanisms of shared depression in romantic couples is limited by the fact that research tends to focus on particular constructs/variables or on a particular cause of depression, particularly aging processes and deteriorated physical health. Several qualitative studies of couples where one or both partners suffered from major depression have attempted to overcome these focal biases and shed light on additional factors associated with the shared experience of depression. For example, in a study by Sharabi and colleagues [90], 33% of all participants spontaneously referred to the emotional toll of depression. Within non-depressed partners, this was the only element referred to by more than a quarter of the cohort. This emotional toll included feelings of internalizing the other’s depression and cycling down with the partner’s deteriorating mood [90]. In the words of one participant: “I feel that we often “trade” or “cycle” feelings of depression between us… I feel my most depressed when my partner is also depressed…”. Some responses made by participants clearly hint at a contagion process. For example: “My unhappy demeanor is picked up upon by my wife, who sometimes internalizes it, allowing my depression to become her depression”.

## 5. Psychosis

Psychotic disorders are “defined by abnormalities in one or more of the following five domains: delusions, hallucinations, disorganized thinking (speech), grossly disorganized or abnormal motor behavior (including catatonia), and negative symptoms” [4] (p. 87). These symptoms may arise from a brief psychotic episode, underlying personality structures, schizophrenia (or similar psychotic disorders), severe depression or mania, an underlying medical condition or ingestion of illegal substances. Psychosis and schizophrenia spectrum disorders have been linked with increased dopamine dysfunction; a disordered neuro-development and a host of psychosocial factors such as childhood adversity, a high sensitivity to stress and social isolation [91].

There is strong evidence supporting the genetic vulnerabilities associated with psychosis, with an emphasis on gene-environment interaction [91]. Studies have also indicated an increased risk among first-level family members. Identical twins have a much higher likelihood of developing a psychotic disorder as opposed to non-identical twins [92], and the familial risk for developing schizophrenia is higher than that for developing psychosis [93]. Other environmental factors which are shared within the familial environment, such as abuse and poor parental attachment, increase the risk of developing psychotic disorders. While the latter are more common when the parent suffers from a severe mental illness, abuse appears to entail an equal level of risk for causing psychosis whether or not the parent is affected by a psychotic disorder [94]. Li and colleagues [94] state that the genetic influence outweighs the effect of the shared environment for siblings, and they support this argument by showing a lower risk (Standardized Incidence Ratio) for spouses to develop psychosis, as compared to instances where there is a genetic relation. However, strikingly, spouses’ risk was still well above that expected in the general population. As mentioned above, common risk factors for psychosis and schizophrenia include the childhood environment, but this does not seem to apply for spouses, who often come from much different backgrounds, thus leaving room for alternative explanations to the shared experience of psychotic symptoms. This is where the possibility of emotional contagion may become relevant.

Assortative mating, which was presented before, was also discussed in the context of psychosis [95]. Despite the argument for spouses both having a shared predisposition for psychosis or schizophrenia, there is a strong indication that the occurrence of psychosis among both members of a romantic dyad reflects more than assortative mating. The increase in concordance between couples’ mental health problems throughout the relationship [96] strongly argues for a contagion-like process in psychosis.

The transmission of psychotic symptoms is defined uniquely in the classic psychiatric and psychoanalytic literature as “Folie a Deux” (“Madness in two”; [97]). In contrast to the contagion of most other psychopathologies, this condition has also been given its own official diagnostic descriptions in the ICD and the DSM. Accordingly, a diagnosis of “Induced Delusional Disorder” may be given if “…Only one of the people suffers from a genuine psychotic disorder; the delusions are induced in the other(s) and usually disappear when the people are separated” [4]. All descriptions of Folie a Deux and similar psychiatric diagnoses do seem to agree that the phenomenon involves the transmission of psychotic or delusional beliefs from a “primary” patient to a “secondary” person with whom they have close emotional links [98]. Despite its uniqueness in psychopathology and individual diagnostic demarcation there has been little research on Folie a Deux [98,99]. Arnone and colleagues [100] conducted a systematic review of the available literature from 1993–2005, finding that the occurrence of Folie a Deux appears to be wider than previously thought, that it most frequently occurs between spouses than siblings, and that separation, the presumably recommended treatment, does not always work, specifically when the “caught” psychosis triggers an already present psychiatric vulnerability in the secondary patient. 

Cases reported in the literature regarding the contagious nature of psychotic symptoms vary with regard to the type of delusions and the relationships between the patients and non-patients. Most frequently, the cases involve two individuals, either siblings or spouses. We note several examples of such case studies. Ghosh [101] describes the transmission of a delusion of being infected with syphilis which was transmitted from patient A to her brother and her brother’s wife (Folie a Trois). In another case study, Magar and Fahy [102] describe two elderly women sharing a hospital ward, with one developing the same psychotic symptoms as the other. More recently, the first published case of Folie a Deux from Zambia [103] described a married couple with five children, where the wife strongly believed herself to be a prophetess and prophesied that the world would soon come to an end. She reportedly passed this belief on to her husband, and they, together with their children had begun praying and fasting, with the children not being allowed to attend school. Generally speaking, the literature shows that psychotic symptoms passed from one patient with psychosis to another appear to exist across the range of common psychotic and delusional experiences, and include somatic [102], persecutory [104], and mystical [105] delusions. They appear to be difficult to treat and have sometimes disastrous consequences [106].

According to the case studies available, susceptibility for contagion in psychosis seems to be based on a number of key factors. There is a general agreement that some a-priori susceptibility to psychosis or psychiatric distress is generally present [100]. In their recent review of shared psychotic disorder in children and young individuals, Vigo and colleagues [107] concluded that “Shared psychotic disorder probably occurs in premorbid predisposed individuals where genetic and environmental factors play an important role in the development of the psychotic episode”. It does also appear that the recipient of psychotic contagion is often younger, more passive, suggestable, less intelligent, has a more dependent personality and suffers from lower self-esteem [99], however much more evidence is needed to support these conclusions. Due to the isolating consequences of psychosis and schizophrenia [108], the enforced closeness of the individual and his/her caregiver or partner also appears to provide a prime breeding ground for the contagion of psychotic symptoms, with most case studies occurring with the infected and infector living together for significant periods of time [101].

## 6. Obsessive Compulsive Disorder (OCD) 

Obsessive compulsive disorder (OCD) is characterized by obsessions and/or compulsions: “Obsessions are recurrent and persistent thoughts, urges, or images that are experienced as intrusive and unwanted, whereas compulsions are repetitive behaviors or mental acts that an individual feels driven to perform in response to an obsession or according to rules that must be applied rigidly” [4] (p. 235). “OCD and related disorders” is now an independent diagnostic category in both the DSM-5 and ICD-11, a diagnostic shift that has been met by controversy [109]. 

The literature regarding OCD transmission or contagion is quite scarce. As in other disorders, one of the more difficult challenges is to separate biological and non-biological processes related to the disorder. OCD is believed to have a strong genetic component [110]. However, knowledge is still lacking regarding specific genes and the role of environmental factors [111], posing a challenge to the understanding of the complex dynamics of symptom transmission. While early onset OCD (in childhood) is thought to be more heritable than late-onset OCD (in adults), this could be related to the greater shared environmental experiences and exposure to parental or sibling OCD symptoms, which are used to calculate heritable risk [110,112]. One of the most extensive studies on the genetic risk of OCD was carried out in Sweden [113]. Interestingly, despite yielding evidence supporting the genetic transmission of OCD, the study also demonstrated that having a child with someone diagnosed with OCD (spouse or partner) increases the partner’s risk of also having OCD [113]. This finding is in concordance with another finding, showing that the spouse of a twin with OCD is more likely to have OCD themselves than the spouse of the twin who does not suffer from OCD [114]. These findings, together with a handful of “Folie a Deux” case studies in OCD [115], indicate that there may be an element of contagion for OCD symptoms. For example, a recent Italian case study [116] described what the authors called “shared OCD” (S-OCD) among a married couple. A 38-year-old OCD patient improved following treatment, and during his symptoms’ remission, his wife started developing the same OCD symptoms. 

An interesting body of the literature stems from OCD family studies. For OCD in families and marital relationships, contagion may be presented in the literature as “family accommodation”. There is a broad range of evidence that families (related and unrelated cohabiters) engage in accommodating behaviors, which replicate OCD rituals in order to reduce the anxiety of the affected relative [117,118]. Family accommodation behaviors include the likes of watching the rituals, waiting for the patient, and modifying the family routine. More important, they also sometimes include the unaffected member taking part in the obsessive activities or rituals [119]. While the levels of family accommodation appear to be generally high in families with a member suffering from OCD, these behaviors do not necessarily equate with their own experiences of OCD symptomology, as studies often show average symptom scores falling well below the validated cutoff points for family members [120]. That being said, there does seem to be a subgroup of relatives who do experience clinically-significant OCD symptoms, and qualify for an OCD diagnosis themselves (e.g., 13% of parents were above the cutoff for OCD when their children were treated for OCD; [121]). Unfortunately, many if not most studies examining family accommodation do not test the partners or family members for their own symptomatology; rather, they simply assess family accommodation in the “unaffected” member. Additionally, when studies do question relatives with regard to their own OCD symptoms, there may be a tendency for the non-OCD individuals to minimize their own experiences. Thus, when those diagnosed with OCD were asked to rate their family members’ OCD symptoms, they were found to have a much more severe presentation [122].

In addition to specific OCD symptomologies, other OCD-related symptoms appear to be transferable. “Harm avoidance” cognitions have been found to be much more common in parent-child dyads with an OCD child patient than when OCD was not present [123]. Additionally, cognitive tendencies such as “anxious foreboding” and an experience of “nervous tension” were more common in relatives of those diagnosed with OCD as compared to relatives of individuals with major depression [124].

It is very difficult to distinguish these shared symptoms and conditions from shared living conditions. For example, parents suffering from OCD have been demonstrated to show a more punitive response to obsessional traits seen in their children, which could, in turn, increase the general stress and anxiety in such family environments [125]. Despite the range of evidence presented thus far, and despite significant evidence indicating that anxiety does tend to spread [22] and is sometimes passed between parents and their children [126], OCD is the least studied of the disorders presented in this paper, in terms of contagion. It is thus a much-neglected field, that warrants further research.

## 7. Discussion 

Emotional contagion and symptom transmission are well-documented phenomena, that have been occupying the minds of mental health professionals and scholars for decades. Thus, it has long been acknowledged that one person’s psychopathology may significantly affect another’s, sometimes to the point of what may appear as full-blown symptom transmission [127]. As psychological science in different fields has clearly shown, no man is an island; rather, emotional distress often occurs within a social context, with those in one’s close environment being affected by one’s symptoms and dysfunction. However, it is quite apparent that while knowledge around emotional contagion is abundant in some areas (e.g., trauma, and somewhat less MDD), it is much scarcer in others (OCD and psychosis). More important, the underlying mechanisms of symptom transmission have yet to be fully understood. In particular, as the field of psychopathology is moving further into attempts to identify transdiagnostic factors, which may explain higher-level phenomena in mental health [128], it seems crucial to assess emotional contagion across disorders, in order to examine shared and differential processes. This paper aimed to meet these goals, by exploring the vast literature on emotional contagion, symptom transmission and related concepts, across four disorders: PTSD, MDD, OCD and psychosis. By examining symptom contagion in each condition separately, we hoped to achieve a better understanding of the mental processes which may underlie psychopathological transmission 

Our comparative analysis argues that in order to better understand emotional contagion and symptom transmission is psychopathology, one needs to take into account four levels: (1) the primary individual coping with mental illness/symptoms, (2) the secondary environment (e.g., spouse, children, therapist, and first responders), (3) unique relational characteristics (e.g., communication patterns between patient and family, enmeshment vs. cutoff), and (4) type of disorder. We believe that the latter level was the one most neglected in existing literature, as we know of no comparative analysis looking at different disorders and revealing the specific dynamics characterizing emotional contagion within them. Figure 1 presents an illustration of the four levels included in our theoretical model.

When exploring the literature, it becomes apparent that the nature of one’s relationships with close others plays a major role in the probability of experiencing contagion in all four disorders. However, the types of interpersonal processes examined in each disorder are quite different. In the trauma literature, a strong emphasis was placed on dysregulation of interpersonal distance. Thus, higher rates of secondary traumatization were often observed among those who employed maladaptive and unbalanced interpersonal patterns (e.g., enmeshment or cutoff from the trauma survivor; [44]). In OCD, however, the interpersonal process of family accommodation seemed to play a role. Perhaps this has to do with the high “visibility” of compulsive rituals that causes family members to adapt their behaviors to align with those of the primary member dealing with the OCD. In MDD, this interpersonal aspect was shown in studies examining attachment cognitions. Finally, in psychosis, this element of dysregulated interpersonal distance is more implicit, perhaps due to a lack of sufficient research in this field. Thus, the “enforced closeness” of others, which stems from the inherent isolation which often characterizes the lives of patients with psychosis, may facilitate the shared experience of psychotic or psychotic-like symptoms. Overall, this comparison between disorders seems to point at certain maladaptive interpersonal patterns, which are characterized by what may be called “dysregulated distance” (too close, too far, or fluctuating between the two). 

An interesting transdiagnostic phenomenon that arises from our analysis has to do with the duality of interpersonal closeness as a process that potentially entails both positive and negative outcomes. In the words of Dekel and colleagues [47], one’s attunement and empathy towards others may serve as a “double-edged sword”. On one hand, closeness and support have been shown to serve as resilience factors in a variety of psychopathologies [129]. However, as was already noted, if these processes are excessive (i.e., reach the point of enmeshment or emotional over-involvement) they may facilitate emotional contagion and place family members and caregivers at risk for psychological distress. This was also shown among parents of children who were hospitalized in a psychiatric ward, where empathy was associated with burden when parental efficacy was low [130]. Interestingly, this dual role of support has also been shown in non-clinical samples, as well as in daily situations (e.g., [131]). This duality seems to exist across disorders and may represent a basic human need to balance between connectedness and aloneness. This notion is in line with classic models and theories in psychology, from Mahler’s separation-individuation theory (1973), through Winnicott’s notion of “the capacity to be alone” (1958), to Blatt’s more modern conceptualizations of Polarities of Experience (2008). Blatt proposes that psychological development is a lifelong personal negotiation between the two fundamental dimensions of relatedness and self-definition. Psychological development, from youth to old age, is a process of synergistic balancing between these two polarities, with most individuals favoring one dimension in particular. Exaggerated emphasis on one developmental line at the expense of the other, however, can lead to a variety of mental disorders. Finally, research has identified some moderators/mediators affecting the outcome of interpersonal closeness (e.g., self-differentiation, [44,132]), but those need to be expanded into a variety of disorders to elicit better intervention and prevention. 

Another important question that our analysis aimed to explore was “what is being transmitted in each disorder?”. While all four disorders examined here each include a wide variety of symptoms (i.e., cognitive, emotional, behavioral), evidence for transmission or contagion are different for various types of symptoms. Looking at MDD, one may notice that much of the literature on contagion refers to the more cognitive aspects of the disorder, including negative beliefs or ideas about oneself, others and the world at large. This is perhaps best exemplified through the notion of “co-rumination” [80], which reflects a shared experience of repetitive negative thoughts between two or more individuals. MDD includes a prominent cognitive aspect [133], with unique negative biases and thinking schemes. These, in turn, may cause one, and one’s environment, to adopt certain ways of thinking, yielding a shared negative experience. In OCD, however, signs of contagion or shared symptoms may be seen particularly in compulsive behaviors [119]. Compulsions are highly visible, as opposed to obsessions, which are more intrapsychic in nature. Thus, compulsive behaviors are noticeable around the house, with family members attempting to accommodate, or even over-accommodate, them. As for psychosis, the literature is mostly based on case studies and thus any inference should be made with caution. Nonetheless, extant literature indicates that delusions are often those that serve as the basis for a shared symptomatic experience. This, again, highlights the contagious nature of psychotic cognitions (i.e., ideas of reference, increased suspicion, and paranoid ideation), which, under certain interpersonal conditions, may be transmitted from one person to another. As noted earlier, the contagion literature related to trauma is considerably vaster compared to all other psychiatric disorders. This is perhaps why there is ample evidence showing that the broadest range of symptom type (emotional, cognitive, behavioral) which is transmitted between the trauma survivor and those surrounding him/her. Negative worldviews, nightmares about the other person’s trauma, avoidance symptoms, shame and anger, have all been found to be experienced as part of what we now term “secondary traumatization”. In this regard, it should also be noted that while the trauma literature naturally shows more psychopathological transmission, recent studies have also begun to show how psychological resilience (i.e., one’s ability to remain stable and maintain well-being following adversity) may also be transmitted within the family system (e.g., [134]). These positive, or salutogenic, aspects of transmission, call for further studies.

It should be noted that one question which seems to “float” at the background of emotional contagion literature has to do with priori vulnerabilities (personality factors, biology, etc.), which may render an individual more susceptible to “catch” others’ symptoms in the first place. Research on these factors is probably much harder, as it requires obtaining pre-contagion data on the individual. In addition, many of these factors are hard to tap or examine, as they may represent very early and/or constitutional processes. Novel research designs, as well as using early archival data for research purposes, may facilitate this line of research, which has the potential to shed important light on contagion processes.

In line with the last comment above, our comparative analysis highlights several methodological issues that need further attention. First, there is clearly a need for much more research in the area of emotional contagion and psychopathology. This is particularly true for non-trauma-related disorders, as PTSD has seen a wide array of studies dealing with vicarious trauma and secondary traumatization. OCD and psychosis are relatively neglected areas, where the published literature is scarce (in the case of OCD) or is heavily based on case studies (in the case of psychosis). The expansion of emotional contagion research into other disorders is also important, as shared psychopathological experiences seem to exist in additional contexts (e.g., eating disorders; [135]). In addition, we strongly urge researchers to adopt novel and more sophisticated methods, including longitudinal studies and application of actor partner analysis (e.g., [86] in the field of MDD), which may shed light on the causal processes where one person develops psychopathology as a result of/in light of another person’s distress. In addition, ecological momentary assessment (EMA) studies, which have become more and more common in psychopathology research (e.g., [136]), may be applied to examine the more subtler, micro-level processes involved in emotional contagion and symptom transmission in various disorders. Another novel methodology, Social Network Analysis, may also prove useful in studying symptom contagion. While this method has been used in studying psychological distress among networks of individuals in some contexts (e.g., internalizing and externalizing problems; [137]) it has been under-utilized in this area of research to date. Finally, we believe that more research is needed in order to understand the effects of symptom contagion on inter-personal relationships. The extant literature shows that those who “catch” another individual’s symptoms may also perceive their relationship with him/her more negatively (e.g., [138]). This may entail important clinical implications. 

Our analysis of emotional contagion is in line with a growing trend in mental health research to look beyond the DSM-led approach of single diagnoses and disorders, and into a more transdiagnostic way of thinking. The Research Domain Criteria [139,140], proposed by the National Institute of Mental Health, suggest looking at mental disorders through a multidimensional lens, which includes information from a variety of areas, from basic biology to behavior, emotion and cognition. This approach inherently places an emphasis on shared mechanisms of distress rather than on exclusive factors contributing to one disorder or another. In line with this approach, we show here how emotional contagion may exist across different disorders, possibly sharing some underlying mechanisms, while differing in others. Nonetheless, this model needs to be further developed, to include more in-depth information about the biological and psychological underpinnings of this somewhat elusive process. 

The integrative review presented here may have several important implications. Theoretically, it is the first attempt, to the best of our knowledge, to explore the phenomenon of emotional contagion in several disorders. When presented side by side, one may draw from data on one disorder to understand the other. Clinically, the main merit of this transdiagnostic approach lies in its potential to identify those who may be at risk of experiencing symptom contagion in general, perhaps due to some pre-existing vulnerability or maladaptive interpersonal patterns. In order to further understand this, future studies need to also include other factors, such as personality traits. Even more important, some of the factors identified here (e.g., emotion regulation, over enmeshment with others, and rumination) are known to be malleable and thus may serve as important targets for psychotherapy (e.g., [141]). Finally, it is important to note that the DSM did acknowledge emotional contagion in two of the four disorders examined here: PTSD (e.g., in the DSM-5 the event criteria include indirect trauma and learning about another’s trauma) and psychosis (i.e., shared psychotic disorder was present in the DSM-5 as a separate disorder, and in the DSM-5, it appears in the section on other specified schizophrenic spectrum and other psychotic disorders, as “delusional symptoms in partner of individual with delusional disorder”). This raises the question of whether it could/should be added for other disorders as well, particularly if some shared underlying mechanisms for contagion do exist, as we suggest here. 

## Figures and Tables

**Figure 1 brainsci-12-00067-f001:**
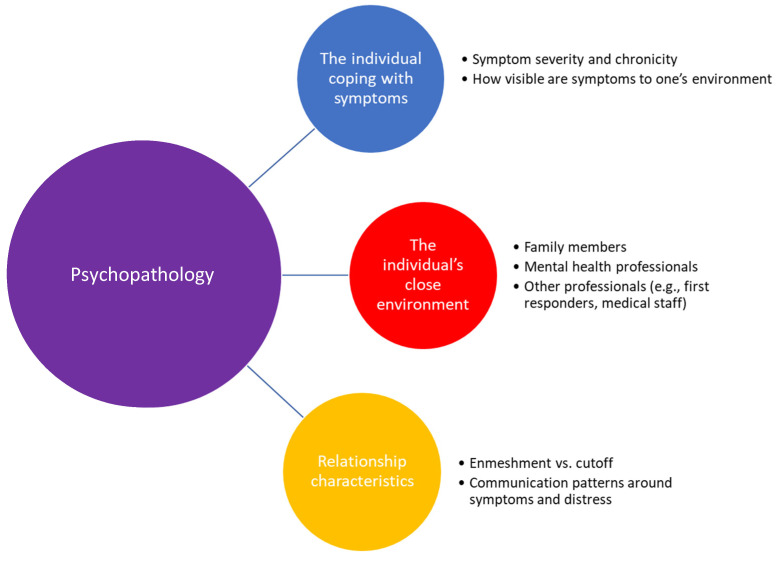
Theoretical model of cross-diagnostic emotional contagion/symptom transmission.

## Data Availability

Not applicable.

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
