# Peer review of "The Contagion of Psychopathology across Different Psychiatric Disorders: A Comparative Theoretical Analysis"

_brainsci, 2021, doi:10.3390/brainsci12010067_

Round 1

Reviewer 1 Report

This manuscript examines the literature on emotional contagion with a particular focus on symptom transmission in four psychiatric disorders: PTSD, Major Depression, OCD, and psychosis. In my opinion, this study represents an interesting contribution. To be suitable for publication in Brain Sciences, however, some issues need to be clarified. The comments below are intended to assist the authors in sharpening a future revision of their paper.

- My main concern regards the focus on emotional contagion, as it seems to me that a narrower focus on symptom contagion would be more appropriate. In fact, the authors mention a transdiagnostic approach but then choose to investigate specific diagnostic categories. If the focus is on emotions, why not looking at it along a continuum (i.e., dimensional view)? Emotion contagion does not only occur in psychopathological individuals but in everybody. Accordingly, examples referring to non-pathological samples are provided. Please expand on the rationale for focusing on psychopathological conditions or focus on symptoms transmission.

- For the same reason, why does Figure 1 include the specific diagnostic entities if it is a cross-diagnostic model? Please revise or comment on that.

- How is the current work examining  a construct that differs from social contagion of the nocebo effect(s)?

-I am aware that the focus is on psychological symptoms but in my opinion the authors cannot neglect a recent meta-analysis conducted on physiological stress contagion that may help understanding the topic of emotion and symptom contagion.

- Related to the previous point, and given the translational nature of the journal, the authors may want to comment on work performed in preclinical samples.

Author Response

  1. We would like to thank the reviewer for the important comment about the emotional contagion framework. We chose emotion contagion as a very general theoretical framework, which has been quite commonly used to explain psychopathology contagion over the years. We certainly agree with the reviewer that it is a very broad umbrella, which includes – among other things – the phenomenon of symptom contagion. That is why, exactly as the reviewer suggested, we often refer to symptom contagion in this paper (please see numerous examples – from the paper’s title and abstract to various sections on pages 1, 4, 7, 12, and many more). That being said, we understand from the reviewer’s comment that perhaps additional reference to symptom contagion is warranted, and that is why we have now further highlighted this specific term throughout our paper (see additions on pages 2, 6, 15).
  2. The reviewer is absolutely right in saying that our main goal was to propose a trans-diagnostic perspective on symptom contagion. However, as we clearly write in the introduction, this perspective is built on first reviewing the literature pertaining to each disorder separately, and only then integrating knowledge from each field in a comparative, cross-diagnostic analysis assessing the shared and differential factors between the 4 conditions. In order to integrate, we first needed to differentiate. That is why we first look at the diagnoses, and then combine bodies of knowledge. We do realize that perhaps this logic was not clear enough, and we now did our best to clarify this is several places throughout the text (please see pages 2, 3, 12). As for the Figure, the same explanation applies to there as well. However, we agree that the theoretical model drawn in the figure can and should follow a more general rationale, and thus we have now changed it according to the reviewer’s suggestion: instead of “type of disorder” and then a list of 4 conditions, it now simply says “psychopathology”.
  3. The idea pertaining to the Nocebo Effect is interesting. We believe that what Nocebo Effects have in common with symptom contagion is that both may be related to negative expectations and negative interpretations, which subsequently may cause significant distress. Nonetheless, we still think that symptom transmission is quite different, as it is a much broader concept, which has been found to result from a variety of factors, as mentioned in our paper.
  4. We thank the reviewer for drawing our attention to the recently published meta-analysis on stress contagion, which adds a different dimension, related to physiology, learning, and non-human models. We now cite this reference in the text (page 3) to highlight the possible basic mechanisms which underlie contagion.
  5. The reviewer is absolutely right in mentioning the importance of work done among pre-clinical samples. In fact, this type of work only goes to show the broader aspects of symptom contagion, that go beyond diagnosed psychopathologies. We now address this line of work (and add new references) on page 3.

Reviewer 2 Report

Thanks for the opportunity to review this paper – it was a very interesting read. This paper reviews how Post-Traumatic Stress Disorder (PTSD), Major Depressive Disorder (MDD), Obsessive-Compulsive Disorder (OCD), and Psychosis might be contagious and proposes what psychological aspects might be transmitted, and how the contagion might happen. This is an original and well-written theoretical piece that I think might be of interest to many scholars.

I find that this conceptualization is presented in a balanced way – and I also agree with the limitations with this perspective – particularly that it might not be the disease themselves that transmits to other people, but feelings of burden, fatigue and burnout. That close family and friends may also suffer when someone has a mental health disease is well-known, but I am not sure that this can be called contagion. Further conceptualization of this should try to delineate this further.

In summary, I do not have specific objections to this paper, but when I read it I had some thoughts that might be helpful:

I wondered about the role of social support in these processes. Perhaps the social support is used so fiercely that the next person gets exhausted, but it may also be that the person with the mental illness is shutting the next person out from their suffering. And this behavior may be the negatively influencing factor for the next person. And how do the assumed contagion processes influence the quality of relationships between people? Do they destroy social bonds or do they strengthen them?

People exposed to traumatic events often feel like they are the only person who have experienced this specific event, and that no one would understand them event (and they might be right). As a result, they may feel disconnected and lonely. I suspect that this may be the case for people with other mental diseases as well. How does the contagion perspective take into account that many of these diseases make people push others away?

I was also reminded by a social network analysis paper, where it was found that risk of depression was higher if people were connected to other depressed people. However, the same was not found for PTSD. Perhaps there are interesting findings or perspectives from the social network literature that could be added if needed. Also I think there are also people (sorry I cant remember the names) who have started working on creating software for analysing social networks of psychopathology – e.g., each person is characterized by their specific individual network of thoughts, emotions and behaviors (aka network analysis), and they are connected to other people (social network analysis) who also have their specific individual networks of thoughts, emotions and behaviors. So there are psychopathological networks within social networks. With data on this it could be possible to look for symptoms (or thoughts/emotions/behaviours) that act as bridge symptoms across individuals. This could be an additional interesting venue for future research on the contagion of mental diseases.

Author Response

  1. We appreciate what the reviewer wrote about the differentiation between symptom contagion and caregiver burden. As he/she rightfully noted, we explicitly said at the beginning of our paper that caregiver burden (and related concepts, such as compassion fatigue) is beyond the scope of our paper, as we believe it involves other processes that are not related directly to symptom contagion.
  2. We appreciate the reviewer’s comment on the role of social support, as well as other inter-personal processes, in symptom contagion. Indeed, both emotional closeness and, alternatively, distance, were found to be associated with higher levels of contagion. We now further emphasize this on page 14 of the revised version. Please also see a detailed discussion of the “double edged” role of support and intimacy in the PTSD section of this paper (page 13). Finally, we appreciate the question about symptom contagion and its effect on relationship quality. We now discuss this on page 15, with the addition of a new reference.
  3. As for the important comment on “pushing others away” – we have revised the paper to include more specific reference to the emotional costs of inter-personal cut-off. Please see page 5-6. Indeed, the paradox is that often, even when people push their close environment away, this very process in fact facilitates contagion. Others may become increasingly worried about the one suffering from symptoms, and in their attempt to assist may themselves become highly stressed and anxious.
  4. We would like to thank the reviewer for the excellent idea to employ social network analysis in the study of symptom contagion. Indeed, this method has been under-utilized in this area to date. We have now added this idea to the “future research” paragraph on page 15.